# Recent Developments in Therapeutic and Nutraceutical Applications of *p*-Methoxycinnamic Acid from Plant Origin

**DOI:** 10.3390/molecules26133827

**Published:** 2021-06-23

**Authors:** Anna Płowuszyńska, Anna Gliszczyńska

**Affiliations:** Department of Chemistry, Wrocław University of Environmental and Life Sciences, Norwida 25, 50-375 Wrocław, Poland; annaplowuszynska@gmail.com

**Keywords:** *p*-methoxycinnamic acid, polyphenols, phenylpropanoids, methoxylated derivatives of cinnamic acid, biological activity

## Abstract

The *p*-methoxycinnamic acid (*p*-MCA) is one of the most studied phenylpropanoids with high importance not only in the wide spectrum of therapeutic activities but also its potential application for the food industry. This natural compound derived from plants exhibits a wide range of biologically useful properties; therefore, during the last two decades it has been extensively tested for therapeutic and nutraceutical applications. This article presents the natural sources of *p*-MCA, its metabolism, pharmacokinetic properties, and safety of its application. The possibilities of using this dietary bioactive compound as a nutraceutical agent that may be used as functional food ingredient playing a vital role in the prevention and treatment of many chronic diseases is also discussed. We present the antidiabetic, anticancer, antimicrobial, hepato-, and neuroprotective activities of *p*-MCA and methods of its lipophilization that have been developed so far to increase its industrial application and bioavailability in the biological systems.

## 1. Introduction

Phenolic acids, due to their broad occurrence in nature and wide spectrum of therapeutic activity, have become the subject of many scientific studies [1]. Among this group of secondary metabolites of plants, *p*-methoxycinnamic acid (*p*-MCA, 4-methoxycinnamic acid, 4MCA) is of particular interest and is one of the most studied methoxy derivatives of cinnamic acid (CA). It is a compound that is commonly found in the plant world and delivered to the body through food products, mainly coffee (*Coffea arabica*) [2], peanuts (*Arachis hypogaea*) [3], cereal plants such as buckwheat (*Fagopyrum esculentum*) [4], and bicolor sorghum (*Sorghum vulgare*) [5]. Rich sources of *p*-MCA also include cereal products such as brown rice grains (*Oryza sativa* L.) [6] and rice bran [7]. This monomethoxy derivative of cinnamic acid has been detected in the leaves of pineapples [8] and roots of a species of banana, *Musa acuminate* [9]. It is also the compound that determines the strong antimicrobial effect of the methanol extract obtained from the seeds of yuzu (*Citrus junos*) fruit against the strains of *Escherichia coli* and *Salmonella enteritidis* bacteria responsible for the majority of gastrointestinal infections in humans [10]. In addition, *p*-MCA is a biologically active ingredient of many other various spices of plants. Its presence has been confirmed in the *Curcuma longa*, kencur galangal (*Kaempferia galanga*) [7] and in the rhizomes of a less known but equally valued plant in southeastern medicine, *Etlingera pavieana* [11]. This compound has been reported as an important component of the herbs of chamomile (*Matricaria chamomilla*) [12], *Anigozanthos preissii* [13], *Dianthus superbus* L. [14], and *Prostanthera rotundifolia* [15] and a constituent of products used in industry like *Hibiscus cannabinus* [16] or Chinese agarwood [17]. Particularly interesting sources of *p*-MCA are endemic plants found in countries of southern Africa, especially *Wachendorfia thyrsiflora* and *Anigozanthos preissii* [13]. So far, *p*-MCA has also been isolated from many wild plants. The most common, naturally occurring sources of this acid include plants from the *Asteraceae*, *Scrophulariaceae*, and *Rutaceae* families. Natural sources of *p*-MCA are presented in Table 1.

## 2. Metabolism of *p*-MCA

The metabolic pathway of *p*-MCA has been studied by Woo following its intravenous and oral administration on a rabbit model [42]. It was observed that for both ways of application, *p*-MCA underwent rapid metabolism and its elimination from the body was clearly dependent on the tract of its delivery. After oral administration of *p*-MCA at a dose of 100 mg/kg to fasted rabbits, its maximum serum concentration of 7.38 mg/100 mL was observed within 1 h. Moreover, during this study, a linear relationship between maximum concentration and dose was observed as well as the differences in the kinetics of the reaction of elimination of *p*-MCA form serum. At a small dose (100 mg/kg), elimination of *p*-MCA was found to proceeded by first-order kinetics, whereas at higher doses (300 and 500 mg/kg), elimination did not follow the first-order kinetics until the amount remaining in the serum was decreased to a concentration of 17 mg/100 mL. Following intravenous injection, the acid was very rapidly metabolized in serum with a half-life 0.4 h. A maximum serum concentration of 41 mg/100 mL was detected three minutes after injection of a dose of 100 mg/kg. For this way of administration, the elimination of *p*-MCA from the serum followed first-order kinetics (the reaction rate was proportional to the substrate concentration and its value decreased with time) and was preceded by an earlier short phase of apparent lateness.

The research conducted by Konishi and coworkers provided more interesting data on the structural specificity of the monocarboxylic acid transporter (MCT) for the transport of *p*-MCA and other phenolic acids by measuring the inhibitory effect on the fluorescein transport across Caco-2 cell monolayers [43]. The amount of fluorescein transported was measured by incubating Caco-2 in the presence or absence of selected phenolic acids at a temperature of 39.9 °C for 40 min. It was observed that all *meta*-substituted methoxy derivatives of benzoic and cinnamic acids showed twice the inhibitory effect or even higher on fluorescein penetration than corresponding *meta*-hydroxy derivatives, suggesting that *meta*-hydroxylation of the substrate would decrease the affinity for MCT. However, this was not observed in the case of *ortho*- and *para*-substituted derivatives and the relative permeation value for *p*-methoxycinnamic acid was estimated as 73.2 ± 7.6%, whereas for *p*-coumaric acid it was 84.1 ± 2.2% [43]. The study of the inhibitory effect on fluorescein transport across Caco-2 monolayers was the initial evaluation of the intestinal absorption and availability of dietary *p*-MCA and other derivatives of aromatic acids. Wen and Walle proved that methylated (poly)phenols are characterized by a five- to eightfold higher oral absorption and higher metabolic stability than their hydroxylated forms [44]. Methylation removes the influence of the highly efficient conjugation pathways limiting the metabolic clearance in the intestinal epithelial cells as well as in the liver.

It has been evaluated that when *p*-methoxycinnamic acid is administrated to humans and rabbits it is oxidized to *p*-methoxybenzoic acid (*p*-MBA), and after subsequent conjugation with glycine and glucuronic acid it is excreted via urine [42]. This compound has been found to be one of five metabolites of ethylhexyl methoxycinnamate (EHMC) [45] most frequently used as UV filters in sunscreens and is a substance rapidly cleared (half-life ≤ 48 min) from human hepatocytes and quickly metabolized [46].

## 3. Therapeutic Activity of *p*-MCA 

Phenolic compounds are of special interest to consumers and food manufacturers due to their wide distribution in the plant kingdom and documented reports on their ability to prevent and reduce the risk of several diseases. The amount of phenolic acid administrated via diet reaches 600–800 mg per day, which is about half of the average dose of all polyphenolic compounds taken every day with food (~1700 mg). Most of this amount (~500–700 mg) are derivatives of cinnamic acid, which can be commonly found in plant foods and beverages [47,48]. Over the last years, the interest of many research groups in the cinnamic acid moiety has notably increased. The number of publications focused on research dedicated to cinnamic acid and its derivatives with the word “cinnamic” in the title in the Scopus database increased from 341 in the year 2003 to 2226 currently in 2021. This proves that natural compounds from this group are increasingly gaining the attention of researchers. In the last two decades, one of the most extensively studied phenylpropanoids has been *p*-methoxycinnamic acid. Many studies in the literature present a wide range of biological properties of this biomolecule as well as its medical applications as an anticancer, antidiabetic, antimicrobial, and neuro- and hepatoprotective agent (Figure 1). We summarized in this review the pharmacological data of *p*-MCA, analyzed its therapeutic outcomes, and reviewed the methods of its lipophilization, which were studied for the increase of its bioavailability in the human body and effectiveness of action so far.

### 3.1. Hepatoprotective Effect of p-MCA

Phenylpropanoids are well known in the literature as the plant metabolites exhibiting hepatoprotective activity [49,50] (Table 2). However, Lee and coworkers studying the relationship between the chemical structure of phenylpropanoids and their hepatoprotective activity have proven that not only the α,β-unsaturated ester moiety, but also the presence of the methoxy group in the *para* position of benzene ring are crucial pharmacophores for this therapeutic property [51]. In vitro tests carried out on rat hepatocytes injured by CCl_4_ showed that *p*-MCA and isoferulic acid exhibit an equipotency effect similar to silybin [52] when used as a supplementary treatment for liver disease at a lower concentration, even 10–50 times lower. Moreover, as a comparison it has been presented that unsubstituted or hydroxy substituted phenylpropanoids possess much weaker hepatoprotective properties. The mechanism of the hepatoprotective activity of *p*-MCA was studied on hepatic levels of glutathione (GSH) and hepatic enzymes related to the GSH redox pathway. While the silybin used as a reference substance at a dose of 50 μM maintained the GSH concentration at a level of half of its concentration in the control sample, *p*-MCA at a concentration of 5 μM significantly preserved the level of total GSH and prevented a decrease in GSH levels by CCl_4_. According to the authors’ observations, *p*-MCA also preserved the activity of glutathione disulfide reductase (GR) and glutathione-S-transferase (GST), which is important for maintaining normal cell homeostasis. 

Similar observations were reported by Fernández-Martinez et al., who studied cinnamic acid derivatives’ activity in preventing experimental CCl_4_-induced liver damage [53]. They also confirmed significant hepatoprotective activity of *p*-MCA in primary cultures of rat hepatocytes among all tested phenylpropanoids. It turned out *p*-MCA exhibited the most protective effect on the plasmatic enzyme activities of alkaline phosphatase (ALP), gamma-glutamyl transpeptidase (γ-GTP), and alanine aminotransferase (ALT) in models of rats intoxicated with CCl_4_. The authors suggest that for this phenomenon responsible different than the antioxidant mechanism and indicate on such pharmacological targets like anti-NF-κB and anti-5-lipooxygenase activities. 

### 3.2. Antidiabetic and Antihyperglycemic Activity of p-MCA

Phenolic acids are known in the literature as natural compounds with potential antidiabetic activity. Molecules belonging to that group have shown during in vitro and in vivo tests that they are able to increase glucose uptake and glycogen synthesis as well as improve glucose and lipid profiles of type 2 diabetes patients [54,55]. Interest in the antidiabetic activity of *p*-MCA has been accelerated in recent years. The insulinotropic and antihyperglycemic properties of *p*-MCA published so far are presented in Table 3.

Adisakwattana et al. proved that, isolated from the rhizomes of *Kaempferia galanga*, *p*-MCA and its ethyl ester act as effective α-glucosidase inhibitors. Using the positive inhibitor 1-deoxynorjirimycin, they determined that the effective dose of *p*-MCA toward α-glucosidase (IC_50_ = 0.044 ± 0.006 mM) is 100-fold lower than the active dose of the compound used as the control (IC_50_ = 5.60 ± 0.42 mM). Among tested cinnamic acid derivatives, *p*-MCA turned out to be the most active inhibitor of α-glucosidase from baker’s yeast as well as *p*-methoxycinnamic acid ethyl ester (IC_50_ = 0.05 ± 0.03 mM). Whereas the first one acted as a noncompetitive inhibitor of α-glucosidase, the latter one was established as a competitive inhibitor. Moreover, obtained results suggest that the presence of a hydroxy or methoxy group at the para position is necessary to enhance α-glucosidase inhibitory activity; however, methoxylated phenylpropanoid at this position is almost 5-fold more active than corresponding hydroxyderivative [56].

The same research group next studied the antihyperglycemic effects of *p*-MCA on plasma glucose and insulin concentration as well as the activities of hepatic glucose-regulating enzymes and hepatic glycogen content in normal and streptozotocin (STZ)-induced diabetic rats. The normal and diabetic rats were fed *p*-MCA at a dose of 40 mg/kg for 4 weeks and it was observed that this acid significantly influenced the activity of the liver enzymes in rats with streptozotocin-induced (STZ) diabetes. While the activity of hexokinase, glucokinase, and phosphofructokinase was significantly raised in diabetic rats after chronic administration of *p*-MCA, in normal ones their levels did not change. It is worth noting that a single administration of this acid to both tested groups of rats reduced plasma glucose concentrations within 30 min. Moreover, an increased glycogen storage in diabetic rats by 80% was estimated. The authors suggest that antihyperglycemic activity is the effect of increased glycolysis and inhibited gluconeogenesis in the liver [57].

Insulinotropic activity of *p*-MCA was further studied in normal and streptozotocin-induced diabetic rats as well as its effect on insulin secretion from perfused rat pancreases and on intracellular calcium in the pancreatic β-cell line [58]. According to the authors, for normal rats, the greatest decrease in glucose concentration and increase in plasma insulin concentration were recorded for animals treated orally with 40 and 100 mg/kg *p*-MCA and these effects were observed 60 min after administration. In diabetic rats, a significant decrease in plasma glucose concentrations was observed, especially at a dose of 100 mg/kg *p*-MCA, and this effect maintained up to 3 h during fasting. One week after induction of diabetes, insulin deficiency was observed in rats, and its concentration was 2.3 times lower than in normal rats. A greater increase in insulin concentration was noted in rats 1h after administration of a 40 mg/kg (730.9 ± 113.9 pg/mL) dose of *p*-MCA than after supplementation with 100 mg/kg (645.2 ± 89.5 pg/mL). These insulin values increased by 2.0 and 1.76 times, respectively, over the course of one hour. In the oral glucose tolerance test, authors confirmed that the plasma glucose concentrations were significantly decreased in both normal and diabetic rats treated with *p*-MCA. Moreover, the insulinotropic activity of *p*-MCA has been reported based on data obtained from perfused rat pancreases. Acid stimulated insulin secretion in the presence of 5.5 mM glucose and enhanced insulin secretion in the presence of 10 mM glucose. These findings suggest that *p*-MCA may find application to patients with diabetes mellitus who have defects in the response of insulin secretion to glucose and nutrient stimulation. The relationship between the structure of cinnamic acid and its derivatives were evaluated by the examination of effects of activity these compounds had on the insulin secretion and intracellular calcium [Ca^2+^]_i_ in the pancreatic β-cell line (INS-1) [59]. It has been confirmed that presence of *p*-methoxy and *m*-hydroxy residues on cinnamic acid are crucial substituents for effective insulin releasing. However, only *p*-MCA was indicated as an agent that can be useful for controlling postprandial hyperglycemia in diabetic patients because *m*-hydroxycinnamic acid undergoes a rapid onset and short duration of action in animal models. In comparison, *p*-MCA is able to produce more prolonged action that maintains the lowering of plasma glucose by increasing the insulin secretion from pancreatic β-cells. 

The confirmed action of *p*-MCA on the regulation of blood glucose and insulin levels prompted the authors of the above-described studies to determine the mechanism by which this acid stimulates insulin secretion [60]. Adisakwattana et al. confirmed that *p*-MCA influences insulin secretion from rat pancreatic β-cells by increasing the influx of [Ca^2+^]_i_ in INS-1 cells via the L-type calcium channels that can be reinforced by synergistic action with the sulfonylurea antidiabetic agent glibenclamide. They also studied the antiglycation properties of cinnamic acid and its derivatives in a bovine serum albumin (BSA)/fructose system [61]. 

Compared to the drugs used in the treatment of diabetes, such as sulfonylurea derivatives, metformin, α-glucosidase inhibitors, or glitazones, *p*-MCA seems to be a safer agent of treating patients with type 2 diabetes. This natural compound did not show in the in vivo tests activity to induce hypoglycemia or secondary insulin secretory insufficiency. This acid not only increases hepatic glycogen storage, but also increases the ability of the impaired pancreas to secrete insulin in the case of diabetes. The inhibition of the formation of advanced protein glycation products observed during *p*-MCA activity tests may indicate its protective effect and therapeutic activity not only in the case of diabetes, but also in other degenerative diseases, including those related to the nervous, skeletal, and circulatory systems.

### 3.3. Anticancer and Chemopreventive Activity of p-MCA

One of the first papers describing the antitumor potential of *p*-MCA was published in 2000 by Hudson et al. and presented the results of an investigation of the activity of the phenol fraction of brown rice on the proliferation and growth of human breast and colon cancer cells (Table 4) [6]. The authors documented the potential inhibitory properties of phenolic acids on colon and breast cancers, including *p*-MCA extracted from cooked rice bran and brown rice. They indicated that phenols might be a good candidate nutraceutical with colon or particularly breast cancer chemopreventive activity. The *p*-MCA obtained from brown rice and applied in a concentration of 50 µM showed the activity to inhibit the proliferation of colon tumor cells of the HT 29 line and human colon cells of the HCEC line, and the obtained results were similar to those observed for the positive controls genistein (30 μM) and tricine (50 µM). It was also reported that *p*-MCA interfered with the colony formation of colon cancer cell line SW480. The mechanism of anticancer action of *p*-MCA toward human colon adenocarcinoma cell lines was evaluated by Gunasekaran et al. They confirmed that this phenylpropanoid induces apoptosis via an increase in caspase 3 and caspase 9 activities, resulting in the release of cytochrome C to cytosol [62]. 

The chemopreventive activity of *p*-MCA has been evaluated in preclinical models of colon cancer against 1,2-dimethylhydrazine-induced (DMH) rat colon carcinogenesis [63]. Supplementation of rats at doses of 20, 40, and 80 mg/kg b.wt/day for 16 weeks resulted in significant inhibition of the formation of aberrant crypt foci and its multiplication and prevented alterations in DMH-induced circulatory and tissue oxidative stress as well as preneoplastic changes. The therapeutic effects were especially visible after supplementation of rats with a dose of 40 mg/kg b.wt. than at the two other tested doses. This amount is very low in comparison with therapeutic doses for other studied so far phenylpropanoids. Moreover, the estimated active anticancer dose of *p*-MCA in rats is equal to 6.4 mg/kg b.wt. in humans [64]. Therefore, authors marked that this dose cannot be directly correlated with the daily dietary/rice intake by human beings. In this case, daily supplementation of *p*-MCA is necessary. The same dose was found to restore the histological changes induced by DMH and modulate the phase I and phase II enzymes. The *p*-MCA reduced the increased DMH activity of phase I enzymes from microsomal fractions, including cytochrome P450, cytochrome b5, cytochrome P4502E1, cytochrome b5 NADPH reductase, and cytochrome P450 NADH reductase compared to groups treated with DMH alone. In the case of phase II enzymes from the cytosolic fractions, exposure to DMH resulted in a significant reduction in the activity of the enzymes glutathione S-transferase (GST), DT-diaphorase (DTD), UDP-glucuronyl transferase (UDPGT), and gamma glutamyl transferase (GGT) compared to the control group [65].

Supplementation with *p*-MCA decreased the size and incidence of tumors in the colon of carcinogen-treated rats [7]. It was shown that this phenylpropanoid acid in a dose of 40 mg/kg b.wt. exhibited ameliorating anticancer effects by altering multiple processes, including proliferation, angiogenesis, invasion, and induction of cell death in DMH-induced rat colon carcinogenesis. However, the molecular mechanism of the action of *p*-MCA still needs more advanced studies as it seems that the putative antioxidant, anti-inflammatory, and anticarcinogenic properties are crucial to this point [63].

### 3.4. Neuroprotective Activity of p-MCA

In 2000, the research group of Kim evaluated the neuroprotective potential of phenylpropanoids isolated from *Scrophularia buergeriana* MIQ roots in vitro. The activities of compounds being the constituents of the extract were defined by the quantitative measure of the release of LDH into the culture media from primary cultures of rat cortical cells injured with glutamate. The obtained results indicate that *p*-methoxycinnamic acid possesses strong neuroprotectant activity. For this acid, the highest percentage of cell viability among tested phenylpropanoids on the level of 78% was recorded at a concentration of 1 μM [66]. 

In the following years, the research had been extended. Using the same culture system, the structure-activity relationship of isolated phenylpropanoids on glutamate-induced neurotoxicity was analyzed. This time, *p*-MCA also exhibited the highest potency of neuroprotective activity in the group of tested phenylpropanoids. The operating effectiveness was especially visible when *p*-MCA was added before and not after glutamate-induction of neurotoxicity. The mechanism of the action of *p*-MCA was next evaluated in the assays performed with two excitotoxins, *N*-methyl-D-aspartic acid (NMDA) and kainic acid (KA), which were used for induction of selective receptor-mediated neurotoxicity in primary cultures of rat cortical cells. It was confirmed that although *p*-MCA showed neuroprotection for both excitotoxins used, it was more effective in protecting cells from NMDA-induced toxicity than from KA-induced neurotoxicity. This finding was supported by separate studies that determined the [Ca^2+^]_i_ and the content of nitric oxide. Moreover, *p*-MCA also inhibited the binding of [propyl-2,3-^3^H]-CGP39653 and [2-^3^H]-glycine to their respective binding sites on rat cortical membranes. However, even at a higher concentration this inhibition was not complete [67]. 

Further research in this area attempted to determine whether phenolic acids and their methoxy derivatives could inhibit the process of amnesia, which is one of the common ailments that occurs in diseases of the nervous system. The research was carried out on mice with amnesia induced in vivo by scopolamine. Among the tested phenylpropanoids, *p*-MCA and isoferulic acid significantly improved the deficit of memory induced by scopolamine, indicating that α,β-unsaturated carboxyl moiety and the *para*-methoxy group in phenylpropanoids might be important elements in their cognition-enhancing activity. The use of these compounds before induction of memory loss completely prevented amnesia, while their administration after induction with scopolamine resulted in the regression of lesions by 60%. In both cases, the determined active doses of acids were 1 mg/kg body weight, and this effect was possible after a five-day supplementation [64].

A lack of evidence for the therapeutic effect of *p*-MCA in chronic memory deficit encouraged Rijal and coworkers to study this acid and its ethyl ester against a chronic model of cognitive dysfunction induced in rats by administering AlCl_3_ (10 mg/kg). Treatment with these compounds showed a significant improvement in spatial memory makers and altered hippocampal AChE activity in rats with cognitive dysfunction. The improvement in memory and cognitive parameters was attributed to the neuronal-protective antioxidant activity of *p*-MCA, and biochemical effects such as AChE regulation only support the neurotherapeutic properties of *p*-MCA in aluminum-induced dementia. Moreover, the authors described the superiority of ethyl *p*-methoxycinnamate (50–100 mg/kg) over *p*-MCA due to better bioavailability of this derivative in the body and better regulation of AChE enzymes [68].

### 3.5. Antimicrobial Activity of p-MCA

Increasing drug resistance of microorganisms has led to the search for new effective agents based on compounds of natural origin. Since it was reported that cinnamic acid exhibits antimicrobial activity, many research groups have started evaluating the relationship between the structure of phenylpropanoids and their microbiological effects. Narasimhan et al. performed in vitro tests evaluating the biological activity of a number of cinnamic acid derivatives against selected strains of microorganisms. They found that while cinnamic acid exhibited a weak antibacterial effect against most of Gram-negative and Gram-positive species of bacteria with minimum inhibitory concentration (MIC) values ranging from 270 μM to 6.75 mM, *p*-MCA showed much higher activity toward the same species with active doses between 50.4 μM and 449 μM. This acid exhibited higher inhibition against fungal species in comparison to bacteria (Table 5) [10,69,70]. *p*-Methoxycinnamic acid turned out to be the most active substance of yuzu (Citrus junos hort. Ex Tanaka) seed extract and is responsible for its antimicrobial effect. *p*-MCA acid and other phenolic compounds are also responsible for the antibacterial and antifungal properties of the extract of roots of *Dendranthema zawadskii* var *latilobum* Kitamura. The 4-methoxylated derivative of cinammic acid showed a moderately broad spectrum of antibacterial activity against both Gram-positive and Gram-negative bacteria [22], whereas its antibacterial activity was significant against cariogenic oral streptococci *Streptococcus mutans* and *Streptococcus sobrinu* [29]. Structurally related to cinnamic acid, phytochemicals were assessed as antimicrobial agents toward *Escherichia coli*, *Staphylococcus aureus*, and *Enterococcus hirae* by Malheiro and coworkers. In the case of *p*-MCA, a tested maximum concentration 25 mM turned out to be too low to establish a MIC and minimum bacterial concentration (MBC). However, its effect on bacterial growth dynamic was clearly visible. *p*-MCA and its tested derivatives were able to affect bacterial growth at 5 mM either increasing doubling time or increasing the duration of the lag phase. The authors also indicated that the mode of action of these phytochemicals depends on the concentration of undissociated acid. It was established that with a lower pH the antimicrobial activity tends to increase since the concentration of undissociated phenolic acids is higher, making them more soluble in cytoplasmic membranes. Moreover, studied molecules were able to inhibit quorum sensing [71]. 

Based on these preliminary results, Cheng et al. studied the activity of 4-dimethylaminocinnamic acid (DCA) and *p*-MCA toward *Chromobacterium violaceum* [72]. Both compounds were identified as potential quorum-sensing (QS) and biofilm inhibitors. It was observed that they reduced the level of *N*-decanoyl-homoserine lactone (C10-HSL) at concentrations of 100 µg/mL and 200 µg/mL, respectively, and inhibited production of virulence factors. They effectively inhibited the expression of *cviI*, leading to a decrease in C10-HSL and eventually a decrease in its binding to the transcription regulatory protein CviR in *C. violaceum*. During the study of the mechanism of antibacterial action of cinnamic acid and *p*-MCA, it was proven that these compounds effectively reduced the production of chitinase and the expression of its regulatory gene *chiA*, which limits the invasiveness and adaptability of the *C. violaceum* growing environment. *p*-MCA increased the sensitivity of biofilms to tobramycin by introducing changes in the structure of the cell membrane. In addition, it has been proven that *p*-MCA reduces the level of cellular metabolites, including l-methionine, ethanolamine, d-proline, *N*-acetyl-l-leucine, l-norleucine, l-valine, and l-ornithine, and increases the levels of l- citrulline and ophthalmate [72].

## 4. Recent Developments in Production of Lipid Derivatives of *p*-MCA 

In 2006, Weber’s team presented results of the lipophilization process of *p*-MCA by the transesterification reaction of methyl *p*-methoxycinnamate with oleic alcohol (*cis*-9-octadecan-1-ol) (Figure 2). The reason for the undertaken research was the fact that long-chain esters of *p*-MCA may be used in the food industry as food additives. During studies, the activity of three commercially available immobilized preparations of lipases, Novozym 435, Lipozyme RM IM, and Lipozyme TL IM, were tested. Enzymatic reactions with these lipases were carried out at 80 °C for 72 h. The highest degree of *p*-MCA conversion of 92% was obtained in the reaction catalyzed by lipase B from *C. antarctica* (Novozym 435). Less activity (60% conversion) was shown by the lipase from *R. miehei* (Lipozyme RM IM), while the lipase from *T. lanuginosus* (Lipozyme TL IM) was essentially not able to catalyze a transesterification reaction [73].

Another process of enzymatic lipophilization of *p*-MCA was described by Lee and coworkers, who carried out the synthesis of its ester derivative in reaction with 2-ethylhexanol (Figure 3). Octyl methoxycinnamate (OMC) was obtained via a direct esterification reaction between the above-mentioned substrates at a temperature of 80 °C using Novozym 435 as a biocatalyst. The highest level of substrate conversion (90%) was obtained in 1 day and this high conversion was favored by the use of nonpolar organic solvents as the reaction medium. The OMC ester obtained as a product is a compound that effectively absorbs UV-B rays, which is why it has been widely used in the cosmetics industry as a nonallergenic sunscreen [74].

The optimization of the enzymatic production of OMC was also carried out a few years later by Kumar. Using the lipase from *Rhizopus oryzae* as a biocatalyst, he obtained OMC with a slightly higher degree of conversion on the level of 91.3% in the reaction of *p*-MCA with 2-ethylhexanol. In the esterification process, cyclooctane was applied as the reaction solvent. The synthesis process, carried out on an enlarged scale at a temperature of 45 °C, resulted in a high yield of octyl 4-methoxycinnamate (88.6%) after 96 h of reaction. The obtained product was then subjected to further tests, which confirmed that it exhibited higher antioxidant activity compared to ascorbic acid and the free form of *p*-MCA acid as well as high antimicrobial activity against certain strains of bacteria, yeasts, and fungi such as: *Escherichia coli*, *Klebsiella pneumonia*, *Salmonella typhi*, *Staphylococcus aureus*, *Candida albicans*, *Aspergillus niger*, *Alternariasolani* and *Fusarium oxysporum* [75]. 

A novel synthetic strategy for the production of a long chain ester of *p*-MCA was presented by Borges’s research group [76]. They obtained new t*rans*-tetradecyl-3-(4-methoxyphenyl)propenoate in the reaction of tetradecyl monomalonate with 4-methoxybenzaldehyde via Knoevenagel condensation (Figure 4). The ester synthesized with a high yield of 95%.

Extensive studies on the chemical and enzymatic production of phenolipids and their biological activities were performed using phospholipids, which are characterized by unique physicochemical properties [77,78,79,80,81,82,83]. In our research group, we developed the biotechnological method of synthesis of phospholipid derivatives of *p*-MCA [84]. In the enzymatic interesterification reaction of egg yolk phosphatidylcholine (PC) with ethyl *p*-methoxycinnamate (E*p*-MCA), catalyzed by Novozym 435 under optimized parameters, after three days *p*-methoxycinnamoylated lysophosphatidylcholine (*p*-MCA-LPC) and *p*-methoxycinnamoylated phosphatidylcholine (*p*-MCA-PC) were obtained in isolated yields of 32% and 3% (*w*/*w*), respectively (Figure 5). Based on the analysis of formed products via thin-layer chromatography (TLC) and high-performance liquid chromatography (HPLC), the pathway of possible changes occurring during the reaction of enzymatic interesterification was also proposed [84]. 

## 5. Conclusions and Future Perspectives

The data summarized above present *p*-MCA acid as a molecule that is widespread in the plant kingdom and possesses noteworthy and useful biological properties. Described literature data provide evidence for the protective role of *p*-MCA in degenerative diseases, leaving no doubt that this dietary compound exhibits very high therapeutic and nutraceutical potential. This biomolecule plays the role of a natural antioxidant and also shows indirect antioxidant activity by inducing endogenous protective enzymes and positive regulatory effects on signaling pathways. Compared to other phenylpropanoids, its prohealth properties are significantly valuable. The main advantage of the industrial application of *p*-MCA is its nontoxicity and metabolizing ability via natural microbes. 

The wide variety of health benefits of *p*-MCA as well as other phenolic acids of plants make them attractive for industry. The intake and the use of these compounds as functional ingredients to enrich foods have been increasing in order to provide health benefits to consumers. Foods rich in phenolic acids have been considered as functional foods and a significant growing interest for this type of product from scientists and industry has been observed since 2000.

Phenolic compounds have been used so far in the food industry mainly as natural preservatives but also as agents for enhancing the organoleptic, color, and sensory qualities. In the case of *p*-MCA it is also worth noting that its antioxidant activity is on the level of 70–80% of antioxidant activity of commercially used synthetic food preservatives like butylated hydroxyanisole (BHA) and butylated hydroxytoluene (BHT). That makes this compound very promising as a new natural preservative, especially when we take into account the toxicity of synthetic antioxidants [10]. In addition to its use as a natural antioxidant and therapeutic agent, it could also find use in the food industry as an antimicrobial agent. Nakazono et al. proved that *p*-MCA exhibits greater antibacterial activity than the preservatives commonly used in food production, such as sodium benzoate, benzoic acid, or gallic acid, against bacteria inhabiting the surfaces of production equipment in the food industry, including *Micrococcus luteus*, *Staphylococcus aureus*, *Escherichia coli* and *Salmonella enteritidis* [10].

Despite such promising results, detailed molecular studies of the biological targets of this acid and other phenolic compounds are still needed using in vitro and in vivo models. Studies geared toward the development of efficient and environmentally friendly methods of production of their lipid derivatives, which, as presented above, may solve the problem of their low bioavailability in the human body, are also needed. This seems to be the future trend in this area as well as the production of valuable formulations based on the use of nanotechnology to overcome problems related to the instability of these compounds, which will be a promising strategy to assure bioavailability and to help overcome problems regarding food processing and ingestion.

## Figures and Tables

**Figure 1 molecules-26-03827-f001:**
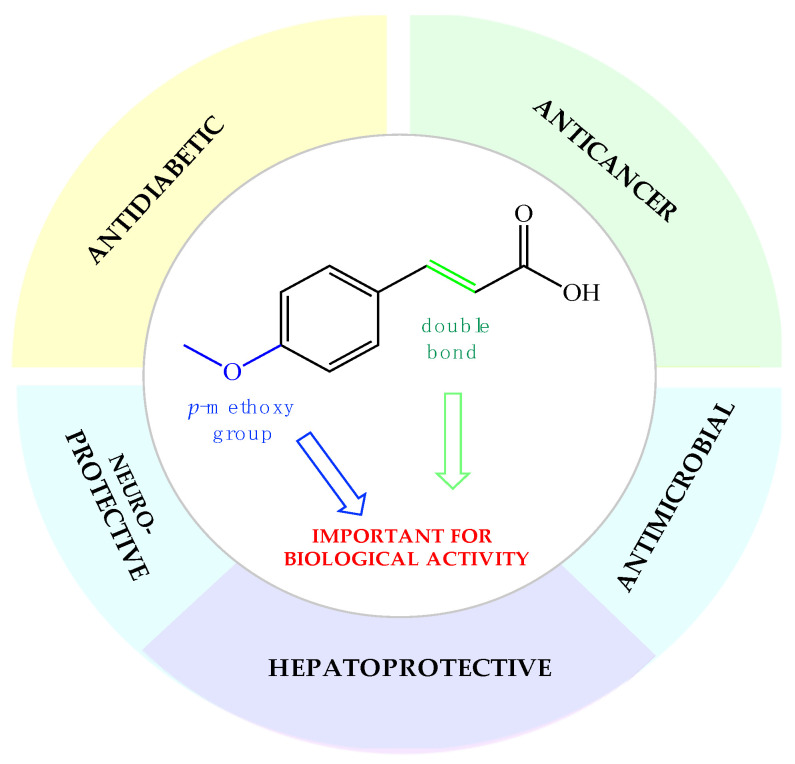
Biological applications of *p*-methoxycinnamic acid.

**Figure 2 molecules-26-03827-f002:**
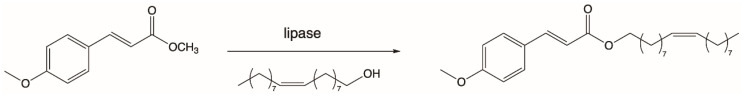
Transesterification of methyl methoxycinnamate with *cis*-9-octadecen-1-ol.

**Figure 3 molecules-26-03827-f003:**
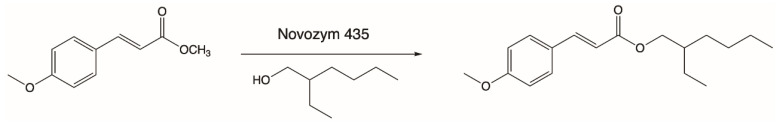
Enzymatic synthesis of octyl methoxycinnamate (OMC).

**Figure 4 molecules-26-03827-f004:**
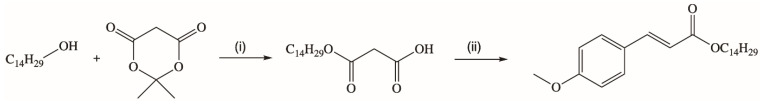
Synthesis of long-chain alkyl *p*-methoxycinnamate. Reaction conditions: (i) toluene, reflux, 4 h (ii) 4-methoxybenzaldehyde, pyridine, β-alanine.

**Figure 5 molecules-26-03827-f005:**
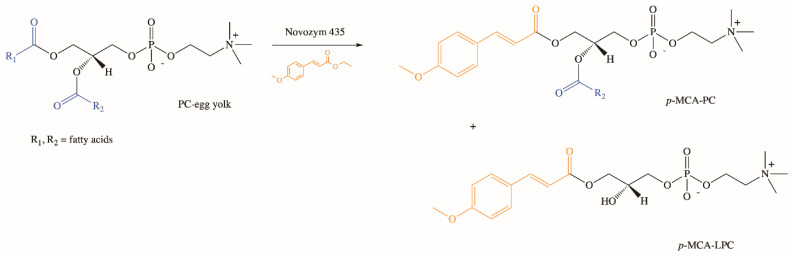
Enzymatic synthesis of phospholipid derivatives of *p*-MCA.

**Table 1 molecules-26-03827-t001:** Natural sources of *p*-methoxycinnamic acid.

Natural Source	Plant Families	Content	Ref.
*Avicennia marina* leaves	*Acanthaceae*	-	[18]
Maples (*Acer* spp.)	*Aceraceae*	-	[19]
*Notopterygium incisum* roots	*Apiaceae*	-	[20]
Carnauba wax, carnauba tree leaves (*Copernicia Cerifera*)	*Arecaceae*	0.3 g/100 g of wax powder	[21]
Chamomile (*Matricaria chamomilla* L.)	*Asteraceae*	-	[12]
Chrysanthemum zawadskii roots (*Dendranthema zawadskii* var. *latilobum*)		36 mg/1.65 kg of fresh roots	[22]
Aerial parts of *Atractylodes macrocephala*		40 mg/15 kg	[23]
*Balanophoru tobiracola* Makino	*Balanophoraceae*	60 mg/8 kg of fresh whole plant (100 g MeOH extract)	[24]
Pineapple leaves (*Ananas comosus*)	*Bromeliaceae*	-	[8]
*Dianthus superbus* L.	*Caryophyllaceae*	-	[14]
*Mallotus conspurcatus*	*Euphorbiaceae*	-	[25]
Peanut hypocotyl and roots *Arachis hypogaea* (most in *Georgia Green*)	*Fabaceae*	1.6 ± 0.7 µg/g of wet *Georgia Green* hypocotyl 17.5 ± 3.7 µg/g of wet *Georgia Green* root	[3]
*Anigozanthos preissii* roots	*Haemodoraceae*	-	[13]
*Wachendorfia thyrsiflora* roots		-	[13]
*Prostanthera rotundifolia* leaves	*Lamiaceae*	94.7 ± 1.0 mg gallic acid equivalent/g DW purified extract	[15]
*Plicosepalus curviflorus* leaves	*Loranthaceae*	350 mg/ 1 kg of leaves	[26]
Kenaf (*Hibiscus cannabinus*)	*Malvaceae*	-	[16]
*Moringa oleifera* leaves	*Moringaceae*	-	[27]
*Musa acuminata* roots	*Musaceae*	-	[9]
*Piper philippinum* stems	*Piperaceae*	0.9–2.1 mg/ 4 kg of dried stems	[28]
Rice bran (*Oryza sativa* L.)	*Poaceae*	-	[7]
*Phyllostachys bambusoides* culms		5.8 mg/5 kg of dried bamboo powder	[29]
Sugarcane juice (*Saccharum officinarum*)		-	[30]
Brown rice (*Oryza sativa* L.)		-	[6]
Sorghum root (*Sorghum vulgare*)		-	[5]
*Polygalae tenuifolia* roots	*Polygalaceae*	64 mg/ 8 kg of whole dried plant	[31]
Buckwheat inflorescences (most in *Fagopyrum esculentum*)	*Polygonaceae*	14.1 ± 7.9 mg/100 g DW of *F. tataricum* 74.7 ± 24.0 mg/100 g DW of *F. esculentum* 17.7 ± 3.3 mg/100 g DW of *F. esculentum* green flowers	[4]
European Columbine leaves (*Aquilegia vularis*)	*Ranunculaceae*	-	[32]
Aerial parts of *Sibiraea angustata*	*Rosaceae*	-	[33]
Green Arabic coffee beans (*Coffea arabica*)	*Rubiaceae*	-	[2]
*Hyptis salzmannii* leaves	*Rutaceae*	0.3 g/1.1 kg of dried leaves powder	[34]
*Murraya euchrestifolia* fruits		60 mg/650 g extract of fruit	[35]
Yuzu seed (*Citrus junos*)		-	[10]
Toddalia root (*Toddalia asiatica*)		-	[36]
*Scrophularia buergeriana*, *S. ningpoensis* roots	*Scrophulariaceae*	0.212 g/4 kg dried powder roots	[37]
Aerial parts of *Buddleja asiatica*		30 mg/12.5 kg of air-dried aerial parts	[38]
*Scrophularia Ningpoensis* roots		8.8 mg/25 kg of dried roots	[39]
*Ailanthus integrifolia* bark	*Simaroubaceae*	7.6 mg/3 kg of dried bark	[40]
Chinese agarwood	*Thymelaeaceae*	-	[17]
*Duranta repens* leaves	*Verbenaceae*	25 mg/17 kg of leaves	[41]
Turmeric (*Curcuma longa* L.)	*Zingiberaceae*	-	[7]
Kencur (*Kaempferia galanga*)		-	[7]
*Etlingera pavieana* rhizomes		-	[11]

“-”—not reported content of *p*-MCA.

**Table 2 molecules-26-03827-t002:** Hepatoprotective effect of *p*-methoxycinnamic acid on injured rat hepatocytes.

	Control	CCl_4_- Intoxicated Rat Hepatocytes	*p*-MCA + CCl_4_	Silybin	Ref.
Concentration (µM)	-	-	5	50	[51]
Total GSH (nmol/mg prot.)	59.74 ± 2.13	9.44 ± 1.02	13.32 ± 0.68	26.51 ± 4.23	
Reduced GSH (nmol/mg prot.)	41.39 ± 1.92	4.08 ± 0.08	10.45 ± 0.83	21.65 ± 1.46	
GSSG/total GSH	0.307	0.621	0.235	0.301	
Glutathione peroxidase (GPx) (nmol/mg protein/min)	1.416 ± 0.018	0.812 ± 0.032	0.831 ± 0.049	0.891 ± 0.024	
Glutathione disulfide reductase (GR) (nmol/mg protein/min)	91.74 ± 11.70	48.73 ± 3.37	68.83 ± 1.24	60.01 ± 3.73	
Glutathione-S-transferase (GST) (nmol/mg protein/min)	1.755 ± 0.135	1.110 ± 0.036	1.466 ± 0.177	1.134 ± 0.086	
Alkaline phosphatase (ALP) (µmol/L min)	107.21 ± 7.36	161.03 ± 12.51	150.33 ± 9.71	-	[53]
Gamma-glutamyl transpeptidase (γ-GTP) (µmol/L min)	8.67 ± 0.24	25.32 ± 1.27	5.86 ± 2.14	-	
Alanine aminotransferase (ALT) (µmol/L min)	33.53 ± 1.23	64.19 ± 1.36	14.12 ± 1.00	-	

**Table 3 molecules-26-03827-t003:** Antihyperglycemic and insulinotropic activity of *p*-MCA.

Research Model	Parameters	Active Dose	Mechanism of Action	Ref
Enzyme	α-Glucosidase	IC_50_ = 0.04 ± 0.01 mM	α-Glucosidase inhibition	[56]
* STPZ-induced rats	Glucose	40 mg/kg	Gluconeogenesis inhibition	[57]
	Insulin		Insulin secretion increase	
	Hexokinase		Hexokinase activity increase	
	Glucokinase		Glucokinase activity increase	
	Phosphofructokinase		Phosphofructokinase activity increase	
	Glucoso-6-phosphatase (G6Pase)		Glucoso-6-phosphatase inhibition	
	Glycogen		Glycogen concentration increase	
	Glucose-6-phosphate (G6P)		Glucose-6-phosphate concentration increase	
* STPZ-induced rats	Insulin	40–100 mg/kg	Insulin secretion increase	[58]
Perfused rat pancreases	Insulin	10–100 µM	Insulin secretion increase	
** INS-1 cell line	Insulin	100 µM	Insulin secretion increase	[59]
Wistar rats	Insulin	5 mg/kg	Insulin secretion increase	
	Glucose		Gluconeogenesis inhibition	
** INS-1 cell line	Insulin	10–100 µM	Insulin secretion [Ca^2+^] influx-dependent increase	[60]
Perfused rat pancreases	Insulin	10 µM	Insulin secretion increase	
Fructose-modified bovine serum albumin	AGE (CML) ***	1 mM	Inhibition AGEs- products	[61]
	Fructosamine		Fructosamine levels reduction	
	Amyloid cross *β* structure		Amyloid cross *β* structure level reduction	
	Thiol group		Thiol groups oxidation reduction	
	Protein carbonyl		Protein carbonyl formation suppression	

* STPZ—streptozotocin; ** INS-1—rat insulinoma cell line; *** AGE—advanced glycation end products; CML—N^ε^-(carboxymethyl)lysine.

**Table 4 molecules-26-03827-t004:** Chemopreventive and cytotoxic activity of *p*-MCA.

Molecule	Research Model	Active Dose	Mechanism of Action	Ref
*p*-MCA	SW480 cell line	200 µM	Anticlonogenic activity	[6]
	HCT-116 cell line	IC_50_ =10.25 ± 0.94 μM	Induction of apoptosis	[62]
	HT-29 cell line	IC_50_ = 11.23 ± 1.02 μM		
	COLO320 cell line	IC_50_ = 12.32 ± 0.81 μM		
	NCM460 cell line	IC_50_ = 137.28 ± 2.37 μM		
	* DMH-induced Wistar rats	40 mg/kg	Induction of apoptosis Anti-inflammatory effect Antiproliferative effect Antiangiogenic effect	[7] [65] [63]

* DMH—1,2-dimethylhydrazine.

**Table 5 molecules-26-03827-t005:** Antimicrobial and antifungal activity of *p*-MCA.

**Gram-Positive Bacteria**		
**Microorganisms**	**Active Dose of *p*-MCA**	**Ref.**
*Bacillus subtilis*	MIC 203 µM	[69]
*Staphylococcus aureus*	MIC 203 μM	[69]
*Micrococcus luteus*	MIC 80 µg/mL	[10]
*Staphylococcus aureus*	MIC 60 µg/mL	[10]
*Bacillus subtilis*	25–100 µg/ disc	[22]
*Staphylococcus aureus*	25–100 µg/ disc	[22]
*Streptococcus mutans*	MIC 352 µg/mL	[29]
*Streptococcus sobrinus*	MIC 470 µg/mL	[29]
*Staphylococcus aureus*	MIC > 25 mM MBC > 25 mM	[71]
*Enterococcus hirae*	MIC > 25 mM MBC > 25 mM	[71]
**Gram-Negative Bacteria**		
**Microorganisms**	**Active Dose of *p*-MCA**	**Ref.**
*Escherichia coli*	MIC 164 µM	[69]
*Escherichia coli*	MIC 50 µg/mL	[10]
*Salmonella enteritidis*	MIC 60 µg/mL	[10]
*Escherichia coli*	25–100 µg/ disc	[22]
*Shigella sonnei*	25–100 µg/ disc	[22]
*Chromobacterium violaceum*	200 μg/mL	[72]
*Escherichia coli*	MIC > 25 mM MBC > 25 mM	[71]
**Fungi**		
**Microorganisms**	**Active Dose of *p*-MCA**	**Ref.**
*Aspergillus niger*	MIC 50.4 µM	[69]
*Candida albicans*	MIC 50.4 µM	[69]

## Data Availability

Data is contained within the article.

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
