# Peer review of "Recent Developments in Therapeutic and Nutraceutical Applications of p-Methoxycinnamic Acid from Plant Origin"

_molecules, 2021, doi:10.3390/molecules26133827_

Round 1
Reviewer 1 Report
The manuscript “Recent development in therapeutic and nutraceutical applications of p-methoxycinnamic acid from plant origin” is a review that summarizes the recent information about this interesting natural compound. The document is well structured, well described and easy to read, there are some minor comments that could be addressed. In the abstract section it is necessary to add the importance of the p-MCA for the food industry In the introduction section and in table 1 the quantity of the p-MCA compound from each specie could be added. In the title it is mentioned the nutraceutical applications of p-MCA, nevertheless in the document there is not really emphasis in this part, the author just mentioned that can be used in the food industry. Please add the information. In the antimicrobial descriptions there is interesting information about the Minimum Inhibitory Concentrations, nevertheless it is not mentioned a mechanisms of action as for the others applications.Author Response
Dr. Matej Sova
Guest Editors of Molecules
Dear Editor in Chief of Molecules,
We would like to thank you for the opportunity to revise our paper on “Recent development in therapeutic and nutraceutical applications of p-methoxycinnamic acid from plant origin” (molecules-1274735). The offered suggestions have been immensely helpful for us. We have included the Reviewers comments and responded to them individually, indicating exactly how we addressed each concern and describing the changes we have made. All changes were highlighted in yellow in revised version of the manuscript.
According to the Reviewer 1 comments:
In the abstract section it is necessary to add the importance of the p-MCA for the food industry. In the introduction section and in table 1 the quantity of the p-MCA compound from each specie could be added. In the title it is mentioned the nutraceutical applications of p-MCA, nevertheless in the document there is not really emphasis in this part, the author just mentioned that can be used in the food industry. Please add the information.
Response: We agree with the Reviewer and we have already added in the abstract and also in the last section (Section 5) of the manuscript information about importance of p-MCA in food industry, its role of nutraceutical agent and future perspectives in this area of research (lines 457-494). We also added in Table 1 the column with the content of this acid in presented plants for these species for which it was possible to find this data in the citated literature.
In the antimicrobial descriptions there is interesting information about the Minimum Inhibitory Concentrations, nevertheless it is not mentioned a mechanism of action as for the others applications.
Response: According to the Reviewer suggestions we added the paragraphs about antibacterial mode of action of p-MCA (Section 3.5. lines 354-375).
According to the Reviewer 2 comments:
Highlight the advantages of using p-MCA as a bioactive compound for human health. Try to indicate p-MCA antibacterial mode of action.
Response: According to the Reviewer suggestions we added the paragraphs about advantages of using of p-MCA as bioactive compound for human health (Section 5) as well as description about is antibacterial mode of action (Section 3.5. lines 354-375).
Include emerging technologies to use p-MCA as therapeutic and nutraceutical compounds. Include future trends to keep working with the obtained data. Try to conclude with a general statement of the most relevant part of this study.
Response: We expanded the Section 5 of manuscript taking into account all suggestions from the Reviewer
According to the Reviewer 3 comments:
General comments-
The section 2-Metabolism of p-MCA can be placed after the biological or therapeutic discussion in section 3. There seems to be more connection of the poor bioavailability or metabolic fate of p-MCA and the lipophilic synthesis for industrial application and improved bioavailability in the biological systems. As discussed in one report, ethyl ester of p-MCA fared better than p-MCA in bioavailability of this derivative in the body and better regulation of AChE enzyme (Ref 70). A similar argument could link the metabolism and lipophilic sections after the therapeutic discussion. Accordingly, lines 107-110 can be revised.
Response: Thank you for the suggestion, however after all we decided to leave used division of sections trying to emphasize in this way that p-MCA, as one of the most commonly widespread phenylpropenoic acids in the world of plants, has been thoroughly tested in terms of its metabolism and therapeutic activity. We understand the Reviewer's intention; however, in the literature the metabolism of free form acid is described. On the other hand, in the section where the therapeutic activity of the acid is presented, in section 3.4 there is the first mention of the better bioavailability of the acid after its administration in the form of an ester. Therefore, the results concerning the production of lipid derivatives of this acid are described in the next subsection.
In recent developments, can the authors include the chemical method for synthesis of lipophilic cinnamic derivatives see https://pubmed.ncbi.nlm.nih.gov/21216503/
Response: We would like to thank you to Reviewer for the indicated paper. We have already added this reference to the section 4 of the manuscript (line 426-435).
Please include reference for the table captions as follows-
Table 2. Hepatoprotective effect of p-methoxycinnamic acid on injured rat hepatocytes. [51,53]
Table 3. Antihyperglycemic and insulinotropic activity of p-MCA. [56-61]
Table 4. Chemopreventive and cytotoxic activity of p-MCA. [62-66]
Table 5. Antimicrobial and antifungal activity of p-MCA [10, 71-75]
Include the DOI for all citations in the references section.
Response: Thank you, we made appropriate changes and included the DOI in the references section.
We would like to express once again our thanks to Reviewers for very valuable comments, which help us to improve our manuscript. We hope that our explanations and corrections are sufficient, and will be accepted.
With kind regards,
Anna Gliszczyńska

Reviewer 2 Report
Dear Author, I reviewed the manuscript (molecules-1274735) entitled Recent Development in Therapeutic and Nutraceutical Applications of p-methoxycinnamic acid from plant origin. This manuscript presents relevant information about novelty applications of p-MCA. However, some sections of the presented data can be improved. For this reason, I considered that this manuscript needs minor changes for being considered for its publication in this journal.
Additional comments.
Highlight the advantages of using p-MCA as a bioactive compound for human health.
Check paragraphs extension in this manuscript.
Try to indicate p-MCA antibacterial mode of action.
Include emerging technologies to use p-MCA as therapeutic and nutraceutical compounds.
Include future trends to keep working with the obtained data.
Try to conclude with a general statement of the most relevant part of this study.
Author Response
Dr. Matej Sova
Guest Editors of Molecules
Dear Editor in Chief of Molecules,
We would like to thank you for the opportunity to revise our paper on “Recent development in therapeutic and nutraceutical applications of p-methoxycinnamic acid from plant origin” (molecules-1274735). The offered suggestions have been immensely helpful for us. We have included the Reviewers comments and responded to them individually, indicating exactly how we addressed each concern and describing the changes we have made. All changes were highlighted in yellow in revised version of the manuscript.
According to the Reviewer 1 comments:
In the abstract section it is necessary to add the importance of the p-MCA for the food industry. In the introduction section and in table 1 the quantity of the p-MCA compound from each specie could be added. In the title it is mentioned the nutraceutical applications of p-MCA, nevertheless in the document there is not really emphasis in this part, the author just mentioned that can be used in the food industry. Please add the information.
Response: We agree with the Reviewer and we have already added in the abstract and also in the last section (Section 5) of the manuscript information about importance of p-MCA in food industry, its role of nutraceutical agent and future perspectives in this area of research (lines 457-494). We also added in Table 1 the column with the content of this acid in presented plants for these species for which it was possible to find this data in the citated literature.
In the antimicrobial descriptions there is interesting information about the Minimum Inhibitory Concentrations, nevertheless it is not mentioned a mechanism of action as for the others applications.
Response: According to the Reviewer suggestions we added the paragraphs about antibacterial mode of action of p-MCA (Section 3.5. lines 354-375).
According to the Reviewer 2 comments:
Highlight the advantages of using p-MCA as a bioactive compound for human health. Try to indicate p-MCA antibacterial mode of action.
Response: According to the Reviewer suggestions we added the paragraphs about advantages of using of p-MCA as bioactive compound for human health (Section 5) as well as description about is antibacterial mode of action (Section 3.5. lines 354-375).
Include emerging technologies to use p-MCA as therapeutic and nutraceutical compounds. Include future trends to keep working with the obtained data. Try to conclude with a general statement of the most relevant part of this study.
Response: We expanded the Section 5 of manuscript taking into account all suggestions from the Reviewer
According to the Reviewer 3 comments:
General comments-
The section 2-Metabolism of p-MCA can be placed after the biological or therapeutic discussion in section 3. There seems to be more connection of the poor bioavailability or metabolic fate of p-MCA and the lipophilic synthesis for industrial application and improved bioavailability in the biological systems. As discussed in one report, ethyl ester of p-MCA fared better than p-MCA in bioavailability of this derivative in the body and better regulation of AChE enzyme (Ref 70). A similar argument could link the metabolism and lipophilic sections after the therapeutic discussion. Accordingly, lines 107-110 can be revised.
Response: Thank you for the suggestion, however after all we decided to leave used division of sections trying to emphasize in this way that p-MCA, as one of the most commonly widespread phenylpropenoic acids in the world of plants, has been thoroughly tested in terms of its metabolism and therapeutic activity. We understand the Reviewer's intention; however, in the literature the metabolism of free form acid is described. On the other hand, in the section where the therapeutic activity of the acid is presented, in section 3.4 there is the first mention of the better bioavailability of the acid after its administration in the form of an ester. Therefore, the results concerning the production of lipid derivatives of this acid are described in the next subsection.
In recent developments, can the authors include the chemical method for synthesis of lipophilic cinnamic derivatives see https://pubmed.ncbi.nlm.nih.gov/21216503/
Response: We would like to thank you to Reviewer for the indicated paper. We have already added this reference to the section 4 of the manuscript (line 426-435).
Please include reference for the table captions as follows-
Table 2. Hepatoprotective effect of p-methoxycinnamic acid on injured rat hepatocytes. [51,53]
Table 3. Antihyperglycemic and insulinotropic activity of p-MCA. [56-61]
Table 4. Chemopreventive and cytotoxic activity of p-MCA. [62-66]
Table 5. Antimicrobial and antifungal activity of p-MCA [10, 71-75]
Include the DOI for all citations in the references section.
Response: Thank you, we made appropriate changes and included the DOI in the references section.
We would like to express once again our thanks to Reviewers for very valuable comments, which help us to improve our manuscript. We hope that our explanations and corrections are sufficient, and will be accepted.
With kind regards,
Anna Gliszczyńska

Reviewer 3 Report
Manuscript ID: molecules-1274735
Title: Recent Development in Therapeutic and Nutraceutical Applications of p-methoxycinnamic acid from plant origin by Anna Gliszczyńska et al.
The authors discuss the natural isolation of p-methoxycinnamic acid (p-MCA) from plants and their therapeutic properties and nutraceutical applications. The reports of p-MCA metabolism, & pharmacokinetics are discussed. Several reports dealing with antidiabetic, anticancer, antimicrobial, hepato- and neuroprotective activities are analysed in-vitro and in-vivo studies. Finally the synthesis methods using lipohilization of p-MCA are discussed.
This manuscript is well written with an appropriate introduction and scientific soundness. This makes it valuable to the scientific community working in the field of natural product like cinnamic acid derivatives.
General comments-
The section 2-Metabolism of p-MCA can be placed after the biological or therapeutic discussion in section 3. There seems to be more connection of the poor bioavailability or metabolic fate of p-MCA and the lipophilic synthesis for industrial application and improved bioavailability in the biological systems. As discussed in one report, ethyl ester of p-MCA fared better than p-MCA in bioavailability of this derivative in the body and better regulation of AChE enzyme (Ref 70). A similar argument could link the metabolism and lipophilic sections after the therapeutic discussion. Accordingly lines 107-110 can be revised.
In recent developments, can the authors include the chemical method for synthesis of lipophilic cinnamic derivatives see https://pubmed.ncbi.nlm.nih.gov/21216503/
Please include reference for the table captions as follows-
Table 2. Hepatoprotective effect of p-methoxycinnamic acid on injured rat hepatocytes. [51,53]
Table 3. Antihyperglycemic and insulinotropic activity of p-MCA.[56-61]
Table 4. Chemopreventive and cytotoxic activity of p-MCA. [62-66]
Table 5. Antimicrobial and antifungal activity of p-MCA [10, 71-75]
Include the DOI for all citations in the references section.
Author Response

(The authors gave the same response as above.)
